# Lithium Enhances Hippocampal Glucose Metabolism in an In Vitro Mice Model of Alzheimer’s Disease

**DOI:** 10.3390/ijms23158733

**Published:** 2022-08-05

**Authors:** Camila Gherardelli, Pedro Cisternas, Nibaldo C. Inestrosa

**Affiliations:** 1Centro de Envejecimiento y Regeneración (CARE-UC), Departamento de Biología Celular y Molecular, Facultad de Ciencias Biológicas, Pontificia Universidad Católica de Chile, Santiago 8331150, Chile; 2Instituto de Ciencias de la Salud, Universidad de O’Higgins, Rancagua 2820000, Chile; 3Centro de Excelencia en Biomedicina de Magallanes (CEBIMA), Universidad de Magallanes, Punta Arenas 6210427, Chile

**Keywords:** lithium, glucose, metabolism, Alzheimer’s disease

## Abstract

Impaired cerebral glucose metabolism is an early event that contributes to the pathogenesis of Alzheimer’s disease (AD). Importantly, restoring glucose availability by pharmacological agents or genetic manipulation has been shown to protect against Aβ toxicity, ameliorate AD pathology, and increase lifespan. Lithium, a therapeutic agent widely used as a treatment for mood disorders, has been shown to attenuate AD pathology and promote glucose metabolism in skeletal muscle. However, despite its widespread use in neuropsychiatric disorders, lithium’s effects on the brain have been poorly characterized. Here we evaluated the effect of lithium on glucose metabolism in hippocampal neurons from wild-type (WT) and APPSwe/PS1ΔE9 (APP/PS1) mice. Our results showed that lithium significantly stimulates glucose uptake and replenishes ATP levels by preferential oxidation of glucose through glycolysis in neurons from WT mice. This increase was also accompanied by a strong increase in glucose transporter 3 (Glut3), the major carrier responsible for glucose uptake in neurons. Similarly, using hippocampal slices from APP-PS1 mice, we demonstrate that lithium increases glucose uptake, glycolytic rate, and the ATP:ADP ratio in a process that also involves the activation of AMPK. Together, our findings indicate that lithium stimulates glucose metabolism and can act as a potential therapeutic agent in AD.

## 1. Introduction

Alzheimer’s disease (AD) is the most prevalent age-related degenerative disease characterized by a dramatic loss of neurons and a disruption in synaptic activity [1,2,3]. The neuropathological hallmarks of AD include the extracellular deposition of amyloid plaques and the accumulation of intracellular neurofibrillary tangles. Although the causative mechanisms of AD remain unknown, accumulating evidence has proposed perturbations of glucose metabolism as a pathological feature in AD [4,5]. Previous studies have shown a drastic and progressive decline in the cerebral metabolic rate for glucose in AD patients [6,7,8]. Importantly, this alteration in glucose hypometabolism can appear years before the onset of symptoms and later correlates with clinical manifestations, suggesting a potential role in disease progression [8]. Although the cause of glucose alterations in AD patients remains unknown, several studies have reported that enhancing brain energy metabolism has strong neuroprotective effects and alleviates symptoms in different AD models [9,10,11,12].

For over seven decades, lithium salts have been used in psychopharmacology, especially for treating bipolar disorder and depression [13,14,15]. Moreover, in recent years, this FDA-approved treatment has also been used to treat neurodegenerative diseases, such as AD [16,17,18]. Indeed, numerous studies led by us and others have described lithium as a drug capable of attenuating AD pathology by decreasing tau phosphorylation and the production of amyloid-β (Aβ), together with a reduction in cognitive decline [19,20,21,22,23,24]. Interestingly, positron emission tomography studies using a radiolabeled glucose analog have also shown a glucose uptake increase in several human and mouse brain regions treated with lithium [25,26]. In contrast, other studies have shown no or negative correlation between lithium and brain metabolism [27,28]. Thus, despite its significant use in therapy, the precise effect of lithium on cerebral glucose metabolism remains obscure. To better understand whether and how lithium carbonate (hereinafter lithium) could help restore metabolic alterations in the brain, we used a primary culture of hippocampal neurons and hippocampal slices from APP-PS1 mice to evaluate changes in glucose metabolism. Our data showed a marked stimulation of glucose uptake and an increase in the activation of the metabolic sensor AMP-activated protein kinase (AMPK) after lithium treatment in both hippocampal neurons and slices. Furthermore, lithium enhanced the expression of the neuronal glucose transporter 3 (Glut3), while promoting glucose oxidation by glycolysis, which positively correlated with the ATP:ADP ratio. In conclusion, our study provides a deeper mechanistic understanding of the effects of lithium on cerebral glucose metabolism and its potential use in AD treatment.

## 2. Results

### 2.1. Lithium Stimulates Glucose Uptake in Hippocampal Neurons

To elucidate whether lithium regulates glucose metabolism, first, we followed the uptake of glucose (^14^C-glucose) in hippocampal neurons treated with either 1-, 10-, or 50-mM lithium over 180 s. Our initial analysis showed no differences in glucose uptake after 1 mM lithium treatment compared to control cells (Figure 1A,B). However, unlike 1 mM, 10 mM lithium-treated neurons exhibited a significant 87% increase in glucose uptake rate at 180 s compared to control. Interestingly, increasing lithium concentration to 50 mM slowed down glucose uptake compared to 10 mM treated neurons, suggesting an inhibitory effect at high lithium concentrations [29,30]. To further characterize the effect of lithium on glucose metabolism, we next incubated neurons with 10 mM lithium (due to the highest observed effect) with cytochalasin B (cyt B), actin polymerization, and a potent glucose uptake inhibitor. As expected, cyt B treatment markedly inhibited glucose uptake, even in the presence of lithium (Figure 1C). However, neurons co-incubated with lithium and cytochalasin E (cyt E), an analog of cyt B that does not inhibit glucose transport, restored the levels of glucose uptake similar to lithium-treated cells. Next, we used the glucose analog 2-deoxy-D-glucose (2DG), which halts glycolysis by inhibiting hexokinase activity [31]. We found similar glucose levels in both lithium + 2DG and 2DG treatments, indicating that the lithium-mediated glucose uptake depends in part on the hexokinase activity. To further study the effect of lithium on glucose transport, we also examined the kinetic properties of glucose uptake in hippocampal neurons. We found that the Michaelis–Menten constant (Km) increased from 9.3 nmol/10^6^ cells to 15.3 nmol/10^6^ cells after lithium treatment, whereas the maximal velocity (Vmax) decreased from 7.5 nmol/10^6^ cells in control cells to 6.9 nmol/10^6^ cells, respectively (Figure 1D).

### 2.2. Lithium Promotes Glycolysis and ATP Synthesis in Hippocampal Neurons

To investigate whether the observed lithium-mediated alterations in glucose uptake generate an effect on glucose metabolic pathways, we first determined the rate of glycolysis in hippocampal neurons. We found a more than 4-fold increase in the rate of glycolysis in lithium-treated cells when compared to control, an effect that was lost after the administration of the potent glycolytic inhibitor sodium dichloroacetate (DCA) (Figure 2A). To further evaluate the effect of lithium on glycolysis, we also measured the activity of hexokinase, which catalyzes the first control step in glycolysis. However, no changes in hexokinase activity were observed between lithium and control neurons, whereas co-incubation of lithium with 2DG strongly decreased its activity (Figure 2B). Once inside the cell, glucose can also be metabolized through the pentose-phosphate pathway (PPP). Thus, next, we measured the rate of ^14^CO_2_ released from [1-^14^C] and [6-^14^C]. We found no significant changes in the rate of glucose metabolized by PPP in lithium-treated neurons compared to control, indicating that lithium promotes glycolysis without affecting the PPP (Figure 2C). To confirm this result, we also measured the activity of glucose-6-phosphate dehydrogenase (G6PDH), the first and rate-limiting enzyme of PPP, and the NADPH/NADP^+^ ratio. Our results show that the G6PDH activity was unaffected by lithium; however, the NADPH/NADP^+^ ratio strongly decreased after lithium administration (Figure 2D,E).

Given the increase in glycolysis, next, we evaluated whether lithium affects ATP synthesis. We noticed a significant 2.6-fold increase in ATP levels in lithium-treated cells relative to the control (Figure 3A). As expected, ATP levels strongly decreased in both lithium-treated and control neurons after administering the ATP synthase inhibitor oligomycin. In contrast to ATP, lithium caused a significant drop in ADP levels, which resulted in a high ATP/ADP ratio (Figure 3B,C). Previous results by our lab and others have described lithium as a pharmacological activator of the cellular energy sensor AMP-activated protein kinase (AMPK) [23,32,33]. Thus, to investigate the effect of lithium on AMPK, we measured the levels of activated AMPK (Thr172 phosphorylation) by ELISA. As expected, the presence of lithium significantly increased the activity of AMPK by approximately 80% compared to control cells, while treatment with compound C (CC), an AMPK inhibitor, abolished its activity (Figure 3D). Interestingly, co-incubating neurons with lithium and CC restored the AMPK activity to control levels, suggesting that lithium can activate AMPK by different mechanisms. Collectively, our results suggest that lithium promotes glucose utilization by glycolysis, which in turn triggers ATP production without affecting the PPP.

### 2.3. Lithium Alters the Genetic Expression of Metabolic Genes

Next, we analyzed the effect of lithium on the expression of several genes related to glucose metabolism in hippocampal neurons. Due to the increase in glycolysis, first we quantified the expression levels of hexokinase and phosphofructokinase-1 (PFK-1), which catalyze the conversion of fructose-6-phosphate and ATP into fructose 1,6-biphosphate and ADP. Our results showed no differences in the expression levels of either of these enzymes in lithium-treated neurons as compared to control (Figure 4A,B). Furthermore, and given the described effects of lithium exposure on Gluts in both skeletal muscle and the brain, we next quantified Glut expression changes [32,33,34]. We found no differences in mRNA expression of Glut1, which mediates the transport of glucose into endothelial cells of the blood-brain barrier (Figure 4C) [35,36]. However, we observed a significant 2.6-fold increase in the expression of the neuronal transporter Glut3 after lithium treatment (Figure 4D). Lithium has been widely described as an activator of the canonical Wnt signaling pathway [37,38,39]. Thus, we also evaluated the expression levels of Cyclin-D1, an essential Wnt target gene. Consistent with previous results, lithium treatment significantly upregulated the expression of Cyclin-D1 when compared to untreated neurons (Figure 4E).

### 2.4. Lithium Enhances Glycolysis and AMPK Activity in Hippocampal Slices of APP/PS1 Mice

Finally, to determine whether lithium can restore the impaired glucose homeostasis in AD, we used hippocampal slices from APP/PS1 mice to follow changes in glucose metabolism. Consistent with our results, the administration of lithium significantly increased glucose uptake in hippocampal slices (Figure 5A). Moreover, cyt B strongly inhibited the uptake of glucose, even in the presence of lithium, whereas no differences were observed after cyt E treatment compared with untreated controls. As expected, the uptake of glucose was also decreased after the administration of 2DG, an effect that was not recovered by lithium. Next, to determine whether lithium can also affect glucose utilization preference by glycolysis in hippocampal slices, we measured the glycolytic flux and hexokinase activity. In accordance with our results, lithium caused a significant 89% increase in the glycolytic rate; however, this increase was reduced compared to cultured hippocampal neurons (Figure 5B). On the other hand, and similar to our results with hippocampal neurons, no changes were observed in the activity of hexokinase after lithium treatment (Figure 5C). Next, we evaluated whether the increase in glycolysis affects the levels of ATP and ADP as well as their ratio in hippocampal slices. Lithium-treated cells showed approximately 40% higher ATP levels relative to untreated cells, whereas the presence of oligomycin strongly lowered the ATP levels (Figure 5D). Similar to hippocampal neurons, lithium also caused a marked reduction in ADP levels in slices than in controls, resulting in a high ATP/ADP ratio (Figure 5E,F). Finally, and to verify the activity of AMPK in hippocampal slices, we evaluated its activation in the presence of lithium. Consistent with our results, we found a substantial increase in the levels of activated AMPK after lithium treatment (Figure 5G). Conversely, samples treated with CC and lithium had a significant reduction in AMPK activation.

## 3. Discussion

Among the pathological hallmarks of AD is the progressive impairment of cerebral metabolism [7,40,41]. This hypometabolic state, characterized mainly by a decline in glucose uptake, occurs several decades before clinical symptoms and correlates with cognitive performance [40,41,42,43]. Importantly, strategies that improve the cerebral metabolic rate for glucose have shown neuroprotective effects [10,44,45,46,47]. Indeed, recently, we reported that the use of andrographolide, a natural compound that enhances the transport of glucose via the Wnt pathway, was able to improve cerebral pathology and cognition in APP/PS1 mice [48]. Lithium has been widely used for decades as a psychotropic agent, especially for treating bipolar disorder and more recently for AD; however, its molecular mechanism remains unclear [22,49]. The data presented in this study shows that lithium was able to stimulate glucose metabolism by inducing the uptake of glucose and by enhancing the glycolytic flux, which restored ATP levels in hippocampal cultures (Figure 6).

Consistent with previous results, lithium caused a significant increase in the uptake of glucose, which once internalized, was preferentially oxidized via glycolysis, resulting in a higher ATP/ADP ratio in hippocampal cultures. A lithium-dependent rise in glucose uptake and glycolysis in the brains of rodents has been described previously [25,50,51]. Similarly, a few studies have also described a positive effect of lithium on glucose metabolism in the human brain. For example, lithium has been shown to increase glucose utilization in several brain areas [26]. Moreover, using positron emission tomography, Kohno and co-workers also observed an increase in cerebral glucose metabolism in the bilateral dorsomedial frontal cortices after 4 weeks of lithium carbonate treatment in healthy subjects, while there was a decrease in the right cerebellum [52]. However, using the same technique, a more recent study found that four years of lithium chloride administration in older patients causes a reduction in glucose metabolism in both hippocampus and cerebellum [27].

Although lithium appears to enhance glycolysis, its effects on ATP have been ambiguous. For example, lithium has been shown to positively or negatively modulate ATP synthesis in different tissues [53,54,55]. These confounding results can be explained by lithium’s ability to bind the Mg^2+^-ATP complex, an association that can act in a tissue-dependent manner [56,57].

Intriguingly, lithium’s treatment did not cause changes in either the PPP flux or in the G6PDH activity; however, we observed a strong reduction in the NADPH/NADP^+^ ratio (Figure 2C–E). To our knowledge, there are no studies assessing the direct effect of lithium on the PPP; nevertheless, its effects on the mitochondrial electron transport chain have been reported previously. Indeed, lithium was shown to promote the activity of three respiratory chain enzymes, complexes I, II, and III in the human frontal cortex [58,59]. Thus, it is possible that lithium altered mitochondrial homeostasis, which caused changes in the NADPH/NADP^+^ ratio [60].

Although the exact molecular mechanisms through which lithium act in the brain remain to be elucidated, several direct targets and intracellular pathways have been described in other tissues. For example, the upregulation of glucose metabolism promoted by lithium has been extensively shown in skeletal muscle and adipose tissue, through a process that seems to involve Glut4 translocation to the plasma membrane and/or prevent its internalization [32,61,62,63,64]. An increase in glucose transport by the upregulation of several other Gluts, including Glut3, has also been reported in the liver and muscle [65]. In addition to glucose transport, lithium has also been shown to inhibit glycogen synthase kinase-3 (GSK-3) and to increase the levels of glycogen synthase, a key enzyme in glycogenesis in skeletal muscle, adipocytes, and hepatocytes [25,64,66,67,68,69]. In the brain, lithium has also been shown to inhibit GSK-3 activity through direct and indirect mechanisms, promoting neurogenesis and producing neuroprotective effects [70,71,72]. Importantly, GSK-3β inhibition has been implicated in the control of glucose metabolism through indirect mechanisms [73]. Among them is AMPK, a central node in the cellular metabolism, which under anabolic stimuli is phosphorylated by GSK-3 on the α-subunit, causing a conformational change that renders AMPK inactive [74,75]. In addition, the inactivation of GSK-3β has also been shown to play a crucial role in the activation of the Wnt signaling pathway [76,77]. Importantly, the activation of this pathway has been shown to enhance the activity of several enzymes that regulate glucose metabolism, including hexokinase and PFK-1, together with the expression of Glut1 and 3 [12,78,79,80]. Thus, the observed increase in both AMPK activity and Cyclin D mRNA expression in our hippocampal cultures could be a result of a GSK-3 inhibition mediated by lithium [23,81]. Notably, GSK-3 has been found to be hyperactivated in the brains of AD patients, further validating the use of lithium to regulate this enzyme in AD [82].

Contrarily, several studies have reported negative or no results after lithium treatment [70]. In one study, Hampel and coworkers tested the effect of lithium in patients with mild AD over a 10-week period [83]. However, the authors did not find significant changes in any of the AD pathological markers tested in plasma or cerebrospinal fluid (CSF). It is worth noting that CSF and plasma do not necessarily reflect changes in the brain, thus these results have to be considered carefully [84]. In addition, no effects were observed in either GSK3 activity, a lithium effector, or cognitive performance. The authors explained their negative results to the short time period of the treatment. Similarly, in a different study, Macdonald and colleagues treated patients with mild to moderate AD with lithium [18]. The authors reported no changes in a questionnaire test used to evaluate cognitive impairment, possibly due to the late stage of the patients. More recently, Forlenza and coworkers tested the long-term effects of lithium in patients with amnestic mild cognitive impairment [85]. Interestingly, they observed a significant decrease in phosphorylated tau, an effect that has been widely validated [86,87,88].

In summary, we showed that lithium restored energy homeostasis by enhancing neuronal glucose uptake and AMPK activity and by replenishing ATP stores through a process that seems to favor glycolysis in hippocampal cultures from both WT and APP/PS1 mice. Mounting evidence supports early glucose alterations as a contributing factor to AD. However, to elucidate the pathogenesis and progression of AD, it is critical to understand the mechanisms that lead to a metabolic imbalance [8,12]. Our study suggests the involvement of Wnt signaling in the regulation of glucose metabolism, yet further analyses are needed to fully characterize the metabolic etiology of AD. Moreover, although our study supports the use of lithium to restore glucose metabolism in vitro, more comprehensive in-vivo longitudinal studies that assess cognitive and behavioral changes are needed to validate the effect of lithium. Similarly, the use of different models that recapitulate other pathological features and represent the vast proportion of AD cases is critical to assessing the potential of novel therapeutic approaches, such as lithium [89,90,91]. In conclusion, these results build toward a bigger understanding of lithium’s therapeutic action and offer a potential intervention to restore glucose metabolism in AD.

## 4. Materials and Methods

### 4.1. Animals

Eight-month-old male APPswe/PS1dE9 (APP/PS1) mice (#034832-JAX) and two-month-old female adult Sprague-Dawley pregnant rats were used in this study (a total of 24 animals, 12 APP/PS1, and 12 WT). Embryo donor females were arbitrarily chosen by the animal facility at Pontificia Universidad Católica de Chile. Animals were maintained at the animal facility of Pontificia Universidad Católica de Chile under a sanity barrier in ventilated racks and in closed colonies. The animals were kept under standard cage density conditions and had access to food and water ad libitum. To avoid animal suffering, the animals were reviewed by technical personnel every day to look for evidence of distress (National Institutes of Health tables of supervision). Animals were randomly allocated to experimental groups. All animal work was approved by the bioethical and biosafety committee of the Faculty of Biological Sciences of Pontificia Universidad Católica de Chile (ethical approval CBB-158/2014). The inclusion/exclusion criteria for this study were the health and body weight of the animals. No animals had to be excluded from this study.

### 4.2. Primary Hippocampal Neuronal Cultures

Primary hippocampal cultures were prepared from rat embryos at embryonic day 17.5, obtained from adult female Sprague-Dawley rats. Briefly, pregnant females were euthanized by decapitation following asphyxiation by isoflurane, a widely used volatile anesthetic agent, and the brains from embryos were removed post-mortem. Hippocampi were then removed and dissected in ice-cold PBS under a dissection microscope, mechanically triturated with scissors, and trypsinized (1%) for 10 min at 37 °C. Pooled hippocampal cells were then further disrupted by resuspension using the pipette and centrifuged at 300× *g* for 1 min. The neuronal cultures were plated on poly-L-lysine coated plates in low glucose Dulbecco’s Modified Eagle’s Medium (DMEM) for 30 min at 37 °C with 5% CO_2_, and the medium was then substituted for Neurobasal medium supplemented with B27 (#17504044, Gibco, Carlsbad, CA, USA), streptomycin (100 U/mL) and penicillin (100 U/mL). Cultures were used after 14 days in vitro. To test for neuronal enrichment (>95% neuronal cells present in cell cultures), we selected cultures randomly and analyzed by immunofluorescence using MAP2 (neuronal marker) and GFAP (glial marker), as described previously [32,92,93].

### 4.3. Hippocampal Slices Preparation

Hippocampal slices were prepared as previously described [94]. Briefly, transverse slices (350 μm) from the dorsal hippocampus were sectioned in cold artificial cerebrospinal fluid (119 mM NaCl, 26.2 mM NaHCO_3_, 2.5 mM KCl, 1 mM NaH_2_PO_4_, 1.3 mM MgCl_2_, 10 mM glucose, 2.5 mM CaCl_2_) and incubated in artificial cerebrospinal fluid for 1 h at 22 °C before use.

### 4.4. Quantitative Real-Time PCR (qRT-PCR)

Total RNA was extracted from hippocampal neurons using TRIzol, following the manufacturer’s protocol. RNA sample concentrations were determined at 260 nm absorbance using a spectrophotometer. RNA integrity was verified on a denaturing agarose gel. The 500 ng of total RNA was used for cDNA synthesis using Superscript IV random primers, according to the manufacturer’s instructions. Quantitative real-time RT–PCR (qRT–PCR) was conducted using SYBR master mix (#4385612, Life Technologies) and 18S mRNA as a control, according to the manufacturer’s instructions, as described previously [95]. As a housekeeping gene, we used cyclophilin, and the values were calculated using the delta Ct and normalized to those of the control gene. Duplicate control reactions for every sample without reverse transcription were included to ensure that the PCR products were not due to amplification of contaminant genomic DNA. We used the following sets of primers: 18S gene, forward 5′-TCAACGAGGAATGCCTAGTAAGC-3′ and reverse 5′-ACAAAGGGCAGGGACGTAGTC-3′; cyclophilin, forward 5′-TGGAGATGAATCTGTAGGAGGAG-3′ and reverse 5′-TACCACATCCATGCCCTCTAGAA-3′; Glut1, forward 5′-ATGGATCCCAGCAGCAAGAAG-3′ and reverse 5′-AGAGACCAAAGCGTGGTGAG-3′; Glut3, forward 5′-GGATCCCTTGTCCTTCTGCTT-3′ and reverse 5′-ACCAGTTCCCAATGCACACA-3′; hexokinase-1, forward 5′-GGATGGGAACTCTCCCCTG-3′ and reverse 5′-GCATACGTGCTGGACCGATA-3′; phosphofructokinase-1, forward 5′-AGGGCCTTGTCATCATTGGG-3′ and reverse 5′-ACTGCTTCCTGCCTTCCATC-3′; Cyclin D1-S: 5′-AAAATGCCAGAGGCGGATGA-3′, Cyclin D1-AS: 5′GCAGTCCGGGTCACACTTG-3′. Data were analyzed using the comparative ΔCT method, as previously described [96].

### 4.5. Glucose Uptake Analysis

As previously described, hippocampal cultures were treated with either Li_2_CO_3_ (hereinafter lithium, 1–50 mM_,_ (#255823, Sigma-Aldrich, St. Louis, MO, USA), cytochalasin B (Cyt B, 20 μM, #14930-96-2, Sigma-Aldrich), 2-deoxy-D-glucose (2-DG, 7 mM, #154-17-6, Sigma-Aldrich) or cytochalasin E (Cyt E, 20 μM, #C2149, Sigma-Aldrich) for 15 min and washed with incubation buffer (15 mM HEPES, 135 mM NaCl, 5 mM KCl, 1.8 mM CaCl_2_, and 0.8 mM MgCl_2_) [92]. Cultures were then incubated for 0–180 s with 1–1.2 μCi ^14^C-glucose (D-[1-^14^C] glucose, #NEC043X, Perkin-Elmer, Waltham, MA, USA) at a final specific activity of 1–3 disintegrations/min/pmol (~1 mCi/mmol). Glucose uptake was then arrested with detention buffer (add 0.2 mM HgCl_2_ to incubation buffer (pH 7.4)) and cultures were lysed in 1 mL of lysis buffer (10 mM Tris-HCL and 0.2% SDS, pH 8.0). In total, 3 mL of scintillating solution (#LS-270, National Diagnostics, Atlanta, GA, USA) were added to the cell lysates and radioactivity was measured using a liquid scintillation counter (TriCarb 2900TR analyzer).

### 4.6. Determination of the Glycolytic Rate

Glycolytic rates were determined as previously described [92,93]. Hippocampal cultures were treated with lithium (10 mM) or sodium dichloroacetate (DCA, 5 mM, # 347795, Sigma-Aldrich, St. Louis, MO, USA) for 15 min. Hippocampal lysates were then placed in tubes containing 5 mM glucose and then washed twice in Krebs–Henseleit solution (11 mM Na_2_HPO_4_, 122 mM NaCl, 3.1 mM KCl, 0.4 mM KH_2_PO_4_, 1.2 mM MgSO_4_, and 1.3 mM CaCl_2_, pH 7.4) containing the appropriate concentration of glucose. After equilibration in 0.5 mL of Hank’s balanced salt solution/glucose at 37 °C for 10 min, 0.5 mL of Hank’s balanced salt solution containing various concentrations of [3-^3^H] glucose (#NET331, Perkin-Elmer, Waltham, MA, USA), was added, with a final specific activity of 1–3 disintegrations/min/pmol (~1 mCi/mmol). Aliquots of 100 μL were then transferred to another tube, placed inside a capped scintillation vial containing 0.5 mL of water, and incubated at 45 °C for 48 h. After this vapor-phase equilibration step, the tube was removed from the vial, a scintillation mixture was added, and the ^3^H_2_O content was measured by counting over a 5-min period. Glycolytic rates were determined by measuring the rate of ^3^H_2_O production from D-[3-^3^H] glucose.

### 4.7. Hexokinase (HK) Activity

Hippocampal cultures were treated with lithium (10 mM) or 2-deoxy-D-glucose (2-DG, 7 mM, #154-17-6, Sigma-Aldrich, St. Louis, MO, USA) for 30 min, and cultures were washed with PBS, treated with trypsin/EDTA, and centrifuged at 500× *g* for 5 min at 4 °C. Lysates were then resuspended in isolation medium (250 mM sucrose, 20 mM HEPES, 10 mM KCl, 1.5 mM MgCl_2_, 1 mM EDTA, 1 mM DTT, 2 mg/mL aprotinin [#A1153, Sigma-Aldrich, St. Louis, MO, USA], 1 mg/mL pepstatin A [#77170, Sigma-Aldrich], and 2 mg/mL leupeptin [(#L8511, Sigma-Aldrich, St. Louis, MO, USA]) at a 1:3 dilution, sonicated at 4 °C, and then centrifuged at 1500× *g* for 5 min at 4 °C. The HK activity of the supernatant was quantified. For the assay, the purified fraction was mixed with the reaction medium (25 mM Tris-HCl, 1 mM DTT, 0.5 mM NADP/Na^+^, 2 mM MgCl_2_, 1 mM ATP, 2 U/mL G6PDH, and 10 mM glucose), and the mixture was incubated at 37 °C for 30 min. The reaction was stopped by the addition of 10% trichloroacetic acid (TCA, #T6399, Sigma-Aldrich, St. Louis, MO, USA), and the generation of NADPH was measured at 340 nm.

### 4.8. Quantification of ATP, ADP and NADPH/NADP^+^

Hippocampal cultures were treated with lithium (10 mM) or oligomycin A (50 μM, #75351, Sigma-Aldrich) for 30 min. The cellular ATP (#A22066, Invitrogen) and ADP (#ab83359, Abcam) levels were measured in slices as previously described using an assay kit according to the manufacturer’s instructions [12,45]. The NADPH/NADP^+^ ratio was measured using a commercial colorimetric assay (#MAK038, Sigma-Aldrich, St. Louis, MO, USA), according to the manufacturer’s instructions, as described previously [97].

### 4.9. AMPKα Activity

Hippocampal cultures were treated with lithium (10 mM) or compound C (CC, 10 μM, #P5499) for 30 min. Active (phospho-T172) AMPK was then measured using the AMPK alpha phospho human ELISA kit (KHO0651, Thermo Fisher, Waltham, MA, USA), according to the manufacturer’s instructions and as described previously [98,99,100].

### 4.10. Pentose Phosphate Pathway Measurements

Glucose oxidation via the PPP was measured as previously described based on the difference in ^14^CO_2_ production from [1-^14^C] glucose (decarboxylated in the 6-phosphogluconate dehydrogenase-catalyzed reaction and in the Krebs cycle) and [6-^14^C] glucose (decarboxylated only in the Krebs cycle) [92]. Hippocampal neurons were treated with lithium (10 mM) for 30 min, washed with ice-cold PBS, and collected by trypsinization. The lysates were then resuspended in O_2_-saturated Krebs–Henseleit buffer and 500 μL of this suspension (~10^6^ cells) were placed in Erlenmeyer flasks with another 0.5 mL of Krebs–Henseleit solution containing 0.5 μCi D-[1-^14^C] glucose or 2 μCi D-[6-^14^C] glucose and 5.5 mM D-glucose (final concentration). The Erlenmeyer flasks were equipped with a central well containing an Eppendorf tube with 500 μL of benzethonium hydroxide. The flasks were flushed with O_2_ for 20 s, sealed with rubber caps, and incubated for 60 min in a 37 °C water bath with shaking. The incubations were stopped by the addition of 0.2 mL of 1.75 M HClO_4_ into the main well, and shaking was continued for another 20 min to facilitate the trapping of ^14^CO_2_ by benzethonium hydroxide. Radioactivity was quantified as previously described by liquid scintillation spectrometry [93,101]. Both D-[1-^14^C] glucose (#NEC043) and D-[1-^16^C] glucose (#NEC045) were purchased from Perkin-Elmer.

### 4.11. Determination of G6PDH Activity

Slices or neurons were washed with PBS, collected by trypsinization (0.25% trypsin-0.2% EDTA (*w*/*v*)), and pelleted. The cultures were then resuspended in isolation medium (250 mM sucrose, 20 mM HEPES, 10 mM KCl, 1.5 mM MgCl_2_, 1 mM EDTA, 1 mM DTT, 2 mg/mL aprotinin, 1 mg/mL pepstatin A, and 2 mg/mL leupeptin) at a 1:3 dilution, sonicated at 4 °C, and centrifuged for 5 min at 1500× *g* at 4 °C. Subsequently, the pellet was discarded, and the supernatant was further separated by centrifugation at 13,000× *g* for 30 min at 4 °C. Finally, the G6PDH activity of the supernatant was quantified in a reaction buffer containing 1 mM ATP and 10 mM glucose-6-phosphate (G6P) for 30 min at 37 °C. The reaction was stopped by the addition of 10% TCA. The generation of NADPH was measured at 340 nm, as described previously [102].

### 4.12. Statistical Analysis

All experiments were performed 3–5 times (biological replicates), with triplicates of each condition in each experimental run. The results are expressed as the mean ± SEM. Data were analyzed by either Student’s *t*-test, one-way or two-way ANOVA, depending on the experimental design followed by a posterior Bonferroni’s test for multiple comparisons. Statistical analysis was performed using the software Prism9 version 9.1.1. (GraphPad, La Jolla, CA, USA). Differences were considered significant when ***** *p* < 0.05, ****** *p* < 0.01 and ******* *p* < 0.001.

## Figures and Tables

**Figure 1 ijms-23-08733-f001:**
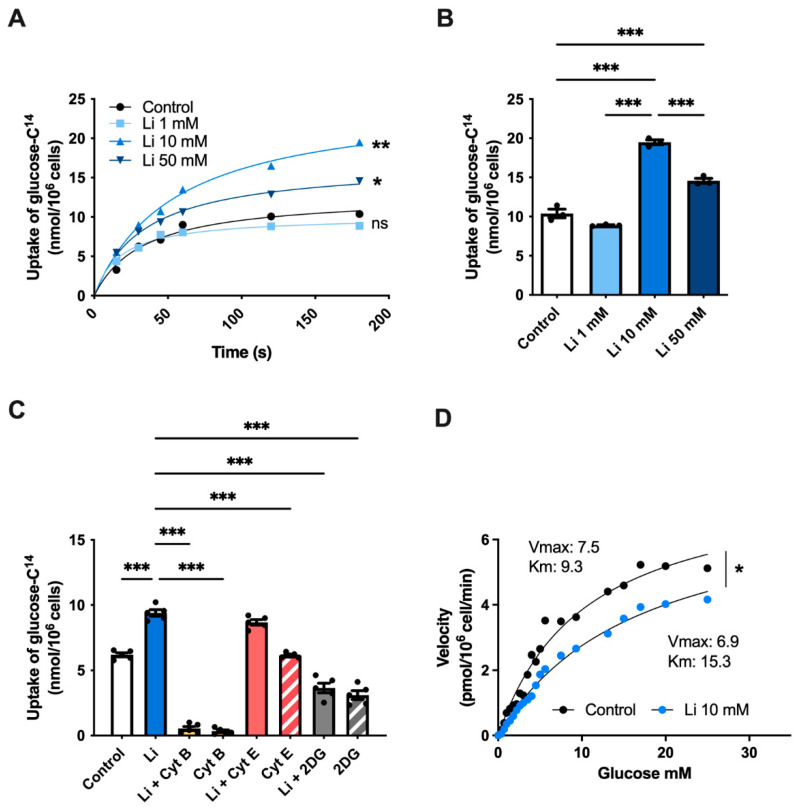
Lithium treatment promotes glucose uptake in hippocampal neurons. (**A**) ^14^C-glucose uptake kinetics curve of hippocampal cultures treated with increasing concentrations of lithium measured over 180 s. (**B**) ^14^C-glucose uptake at 180 s of (**A**). (**C**) ^14^C-glucose uptake competition assay. Hippocampal cultures were treated with 10-mM lithium in the absence or presence of cytochalasin B (Cyt B), cytochalasin E (Cyt E) or 2-deoxy-D-glucose (2DG). (**D**) Michaelis–Menten kinetics of hippocampal neurons incubated with 10-mM lithium and ^14^C-glucose for 30 min. Data were analyzed by nonlinear regression and the Michaelis–Menten equation was used to determine kinetic parameters V_max_ and K_m_. Data plotted as means ± SEM (n ≥ 3 independent cell culture preparations). ***** *p* < 0.05, ****** *p* < 0.01, and ******* *p* < 0.001 were determined by one-way ANOVA (in (**B**,**C**)) or two-way ANOVA (in (**A**,**D**)) followed by Bonferroni’s post hoc test for multiple comparisons; ns, not significant.

**Figure 2 ijms-23-08733-f002:**
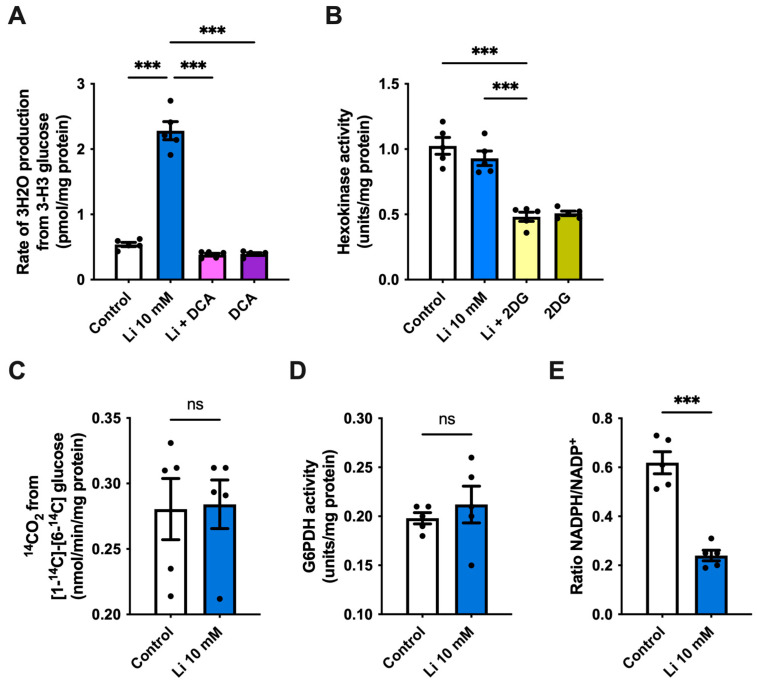
Lithium alters hippocampal bioenergetic status. (**A**) Glycolytic flux of hippocampal neurons treated with 10-mM lithium in the presence or absence of sodium dichloroacetate (DCA) (**B**) Hexokinase activity in hippocampal neurons treated with 10-mM lithium or 2-deoxy-D-glucose (2DG). (**C**) Pentose phosphate flux of hippocampal neurons treated with 10-mM lithium. (**D**) Glucose-6-phosphate dehydrogenase (G6PDH) activity in hippocampal lysates treated with 10 mM lithium. (**E**) NADPH/NADP^+^ ratio of hippocampal neurons treated with 10-mM lithium. Data plotted as means ± SEM (n = 5 independent cell culture preparations). ******* *p* < 0.001 were determined by one-way ANOVA (in (**A**,**B**)) followed by Bonferroni’s post hoc test for multiple comparisons or Student’s *t*-test (in (**C**–**E**)); ns, not significant.

**Figure 3 ijms-23-08733-f003:**
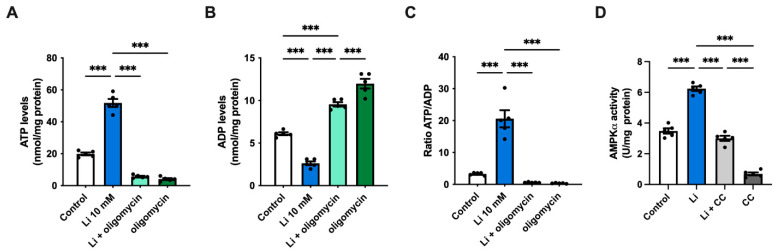
Lithium increases ATP levels and triggers the activation of AMPKα. (**A**–**C**) ATP and ADP levels, and ATP:ADP ratio of hippocampal lysates treated with 10-mM lithium in the presence or absence of oligomycin. (**D**) AMPKα activity in hippocampal neurons treated with either 10-mM lithium or the AMPK inhibitor compound C (CC). Data plotted as means ± SEM (n = 5 independent cell culture preparations). ******* *p* < 0.001 were determined by one-way ANOVA followed by Bonferroni’s post hoc test for multiple comparisons.

**Figure 4 ijms-23-08733-f004:**
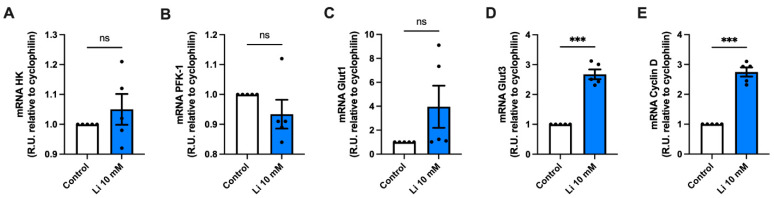
Lithium enhances the expression of neuronal glucose transporter 3**.** (**A**–**E**) mRNA levels of hexokinase (HK), phosphofructokinase-1 (PFK-1), glucose transporter 1 (Glut1), Glut3, and Cyclin D1 (relative to cyclophilin) from hippocampal lysates, measured by qPCR. Data plotted as means ± SEM (n ≥ 4 independent cell culture preparations). ******* *p* < 0.001 was determined by Student’s *t*-test; ns, not significant.

**Figure 5 ijms-23-08733-f005:**
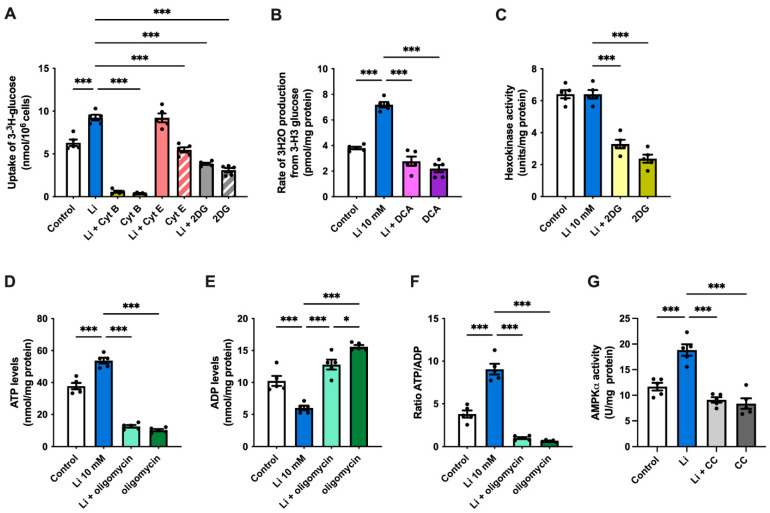
Lithium increases glucose uptake, ATP levels and glycolytic rate in hippocampal slices from APP/PS1 mice (**A**) Effect of 10-mM lithium on ^14^C-glucose uptake in hippocampal slices co-treated with either cytochalasin B (Cyt B), cytochalasin E (Cyt E) or 2-deoxy-D-glucose (2DG). (**B**) glycolytic rate of hippocampal slices from APP/PS1 mice treated with 10-mM lithium in the presence or absence of sodium dichloroacetate (DCA) (**C**) Hexokinase activity in APP/PS1 slices treated with 10-mM lithium or 2-deoxy-D-glucose (2DG). (**D**–**F**) ATP and ADP levels, and ATP:ADP ratio in hippocampal slices treated with 10-mM lithium in the presence or absence of oligomycin. (**G**) AMPKα activity in hippocampal neurons treated with either 10 mM lithium or compound C (CC). Data plotted as means ± SEM (n = 5 independent cell culture preparations). ***** *p* < 0.05 and ******* *p* < 0.001 were determined by one-way ANOVA followed by Bonferroni’s post hoc test for multiple comparisons.

**Figure 6 ijms-23-08733-f006:**
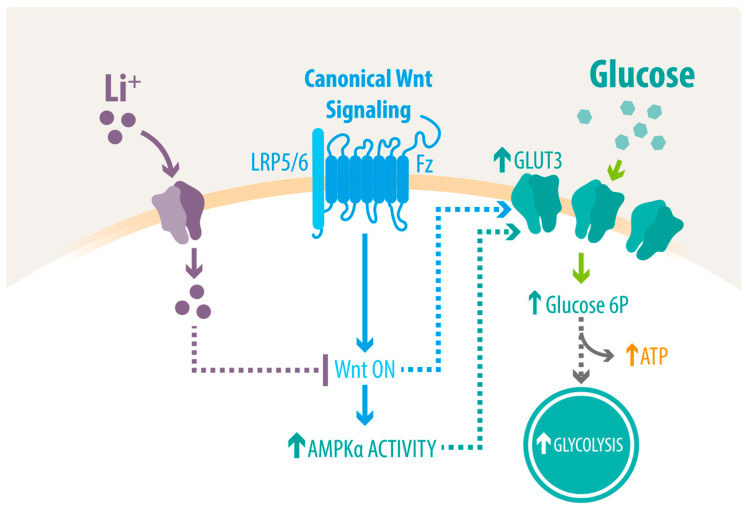
Schematic model for the action mechanism of lithium. Once internalized, lithium results in the activation of the canonical Wnt signaling pathway, promoting AMPK activation. Both AMPK and the Wnt pathway activation can trigger the upregulation of neuronal Glut3, resulting in glucose metabolism restoration by the stimulation of glucose uptake, increased glycolysis, and elevated ATP levels. Dashed lines indicate the participation of several proteins.

## Data Availability

Not applicable.

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
