# Peer review of "Lithium Enhances Hippocampal Glucose Metabolism in an In Vitro Mice Model of Alzheimer’s Disease"

_ijms, 2022, doi:10.3390/ijms23158733_

Round 1
Reviewer 1 Report
This study by Gherardelli et al showed interesting results lithium enhances hippocampal glucose metabolism in mice model of Alzheimer's disease. The paper was well-written, and each parameter tested was well-described and discussed. However, there are some points that need to be addressed:
Major comments:
(1) As the authors discussed the mechanism in Figure 6. It shows the involvement of GSK-3 pathway and WNT pathway. And these are from references. I was confused at first as I thought the authors confirmed this in their study. Is there a way to confirm this in your samples? If not, maybe figure 6 can be edited in such a way that shows the information on these signalling pathways are from previous studies- Just to avoid misleading figure representation.
(2) How many mice were used in the study? In Figure 5, n=5 independent cell culture preparations were used but does it not clearly say how many mice were used- I suggest adding to the figure legend or to Materials and Methods (4.1).
(3) What are the limitations of this study?
(4) I find the conclusion a little weak. Combining the limitation of the study and future prospective study may help strengthen the conclusion and significance of the study.
Minor comments:
(5) Use of however in line 45- suggestion- “Moreover”- (It seems that the use of lithium was additional not contradictory)
(6) Lithium chemical symbol in line 55 is not lithium- or the author meant Li2CO3 could help restore metabolic alteration in the brain – a little confusing and can be misleading
(7) Li in line 195- change to lithium to be consistent
Reviewer 2 Report
In this manuscript, the authors assess the role of lithium on glucose metabolism using primary hippocampal culture and brain slices from APP/PS1 mice. The work is interesting, and the data are strong and well-presented. To improve the manuscript further, the following should be address
The title is misleading. From reading it, one would assume that the work has been done in vivo only to find out that it has been done entirely in vitro.
The authors should discuss the work by Forlenza et al., 2014, which appears inconsistent with the data presented here.
Fig. 6 enriches the manuscript but it, too, is misleading. As it is, one would assume that all the interactions described in the figure are the results of the work presented in this manuscript, which is not obviously the case.
